# Neural networks adaptive time-varying formation control for quadrotor unmanned aerial vehicles with finite-time prescribed performance

1st Yizhe Cao
*School of mechanical engineering*
*Northwestern Polytechnical University*
Xi'an, China
caoyizhe@mail.nwpu.edu.cn

2nd Shulin Song
*Basic Education College*
*Aviation University Air Force*
Changchun, China
295099633@qq.com

3rd Jiahe Huo
*School of mechanical engineering*
*Northwestern Polytechnical University*
Xi'an, China
huojiahe55@gmail.com

4th Dianbiao Dong
*School of mechanical engineering*
*Northwestern Polytechnical University*
Xi'an, China
dongdianbiao@mail.nwpu.edu.cn

5th Dengxiu Yu
*School of Artificial Intelligence, OPtics and ElectroNics*
*Northwestern Polytechnical University*
Xi'an, China
yudengxiu@126.com

*Abstract*—The time-varying formation control problem for quadrotor unmanned aerial vehicles with finite-time prescribed performance subject to nonlinear disturbances is investigated in this paper. To address the unknown nonlinear disturbances, neural networks are introduced. In addition, a novel performance function is formulated specifically for finite-time control. A time-varying formation control strategy for quadrotor unmanned aerial vehicles, which aims to confine the sliding mode error within a prescribed region during a finite time interval, is proposed in this study. This method guarantees both finite-time stability in the closed-loop system and error convergence towards the desired boundary. Subsequently, simulations are performed to confirm the effectiveness and feasibility of the proposed algorithm.

*Index Terms*—quadcopter unmanned aerial vehicles, prescribed performance control, neural networks, finite-time control

## I. INTRODUCTION

Due to the use in high-rise building fire rescue, map generation, wildlife monitoring [1], aerial photography [2], plant protection [3], quadcopter unmanned aerial vehicles (QUAVs) have attracted a lot of attention recently. Researchers have used neural networks (NNs) for the control of QUAVs and have shown significant results in related studies. For example, [4] proposed an innovative fault-tolerant control scheme by integrating virtual estimating algorithms with NNs. [5] presented an learning control scheme for QUAVs with unknown moment of inertia. Meanwhile, researchers have suggested a number of control mechanisms for QUAVs, including adaptive robust control ( [6]) and backstepping control ( [7]). Among these techniques, sliding mode control has been extensively researched and proven to be successful.

Dianbiao Dong is corresponding author.

In control systems, it is common to encounter unexpected nonlinear disturbance terms due to external disturbances and modeling errors. These disturbances can deteriorate the control performance of QUAVs, leading to unstable closed-loop systems or fluctuations in errors. To address this issue, researchers have focused on prescribed performance control (PPC), which aims to confine errors within prescribed regions. By confining the errors, PPC can improve stability and reduce fluctuations in the control system, ultimately enhancing the overall performance of QUAVs. Prescribed performance functions for controlling QUAV have been developed through extensive research, as evidenced by notable studies [8]–[12]. These advancements have resulted in significant improvements in QUAV performance.

However, the studies mentioned above have not yet solved the finite-time controllability problem of QUAVs. Moreover, in real-world scenarios, the convergence speed is a vital indicator of how well a control system is functioning. To assess the performance of QUAVs, several researchers have proposed new error definitions and new observers, such as those presented in [13] and [14]. Further information can be found in studies like [15]–[20], which have presented a range of finite-time stabilization methods including adaptive sliding mode control, Lyapunov stability theory, and finite-time stability theorems. For instance, [15] presented a design method for adaptive sliding mode control (SMC) to stabilize QUAV systems in finite time under parametric uncertainty. [16] proposed a backstepping strategy for finite-time convergence of switching surfaces. In [17], the finite-time stability theorem is used as part of the design of the control such that each QUAV has sufficient tracking performance in a finite amount of time.

In addition, the field of QUAVs has seen significant advancements with the incorporation of finite-time prescribed

performance concepts, see [21]–[26]. For instance, in [21], a fixed-time control scheme was proposed by combining PPC, backstepping design methods, and command filtering. On the other hand, a finite-time command filter is applied in [22] to approximate the derivative of the virtual control law. Moreover, [23] investigated a new finite-time control in the case of executor faults.

Based on the above research, current finite-time prescribed performance research on QUAV focuses on single QUAV control rather than swarm control. Therefore, this research aims to propose a QUAVs time-varying formation control strategy with unkown nonlinear disturbances by utilizing a novel finite-time performance function (FTPF). This research makes the following main contributions:

1) For the time-varying formation control of QUAVs with nonlinear disturbances, this paper proposes a finite-time prescribed performance control strategy by limiting the error of QUAVs within a prescribed region and introducing a neural network to solve the nonlinear disturbance.

2) This paper designs a new finite-time prescribed performance function. Compared with other finite-time prescribed performance functions or prescribed performance functions [26], [27], it has the advantages of small error and settable convergence time.

## II. BACKGROUND

### A. Graph Theorem

$G = \{V, E\}$ is a graph, where $V = \{1, \cdots, N\}$ specifies the vertex set. The number of intelligent agents is $N$, and the edge set is $E = \{(i, j)\}$. Using this architecture, we can express the relationship between the intelligent agents using a matrix. The adjacency matrix is defined as $A = [a_{ij}] \in R^{N \times N}$, where $a_{ij} \neq 0$ if $(i, j) \in E$, and $a_{ij} = 0$ when $(i, j) \notin E$. If $a_{ij} > 0$, then $j$ is designated as a neighbor of $i$, meaning that the two intelligent agents will be able to communicate. $D = (d_i)$ represents the degree matrix and $d_i = \sum_{j=1}^{N} a_{ij}$. $\bar{A} = [\bar{a}_{ij}] \in R^{N \times N}$ represents the normalization adjacency matrix, and $\bar{A}$ is defined as follows:

$$\bar{a}_{ij} = \begin{cases} a_{ij}/d_i, & d_i \neq 0 \\ a_{ij}, & d_i = 0. \end{cases} \tag{1}$$

For the diagonal matrix $\bar{D} = \text{diag}(\bar{d}_i)$, it is defined as below:

$$\bar{d}_i = \begin{cases} 1, & d_i \neq 0 \\ 0, & d_i = 0. \end{cases} \tag{2}$$

The Laplacian matrix $L$ is is obtained by subtracting $\bar{D}$ form $\bar{A}$.

Each of $1, \cdots, N$ stands for a different class of intelligent agent. The global leader when $d_i = 0$ is $i$, which denotes that it doesn't receive information from other intelligent agents.

And we set $B = \text{diag}(b_i)$ as follows:

$$b_i = \begin{cases} 0, & d_i \neq 0 \\ 1, & d_i = 0, \end{cases} \tag{3}$$

therefore, we can then figure out that $\bar{D} + B = I_N$.

**Remark 1:** From the above definition we can see that when $i$ isn't the global leader, $\bar{d}_i \neq 0$ and $b_i = 0$. If $i$ is then $\bar{d}_i = 0$ and $b_i \neq 0$.

### B. Finite-Time Stability Theorem

**Definition 1 [28]:** Contemplate the creation of a system:

$$\dot{x} = f(x, t), \quad f(0, t) = 0, \tag{4}$$

there is always a settling time $T$ for any initial condition $x_0$ at any initial time $t_0$, and the system is stable when $t$ is greater than $T$, $\|x(t)\|$ is less than a very tiny value $\chi$.

**Lemma 1 [29]:** A positive definite function $V(x)$ which is smooth and some positive factor $\vartheta > 0, 0 < \epsilon < 1$ and $s > 0$, such that $\dot{V}(x) \leq -\vartheta V^\epsilon(x) + s, \quad t \geq 0$ is established, thus the system described above is stable.

**Lemma 2 [30]:** The following inequality is valid:

$$\jmath^\imath \ell^\varrho \nu \leq \sigma \jmath^{\imath+\varrho} + \frac{\varrho}{\imath + \varrho} \left[ \frac{\imath}{\sigma(\imath + \varrho)} \right]^{\frac{\imath}{\varrho}} \ell^{\imath+\varrho} \nu^{\frac{\imath+\varrho}{\lambda}}, \tag{5}$$

where $\jmath \geq 0, \imath \geq 0, \ell \geq 0, \varrho \geq 0, \nu \geq 0, \sigma > 0$.

### C. Finite-Time Prescribed Performance Function

**Definition 2:** $v(t)$ represents the finite-time performance function (FTPF) satisfying the conditions: (1) Regardless of the any time $t$, $v(t) > 0$ (2) Regardless of the any time $t$, $\dot{v}(t) \leq 0$ (3) When $t = T_f$, the function is continuous and the value of the function is $v_{T_f}$, where $v_{T_f}$ represents a tiny positive constant and $T_f$ expresses the prescribed time.

On the basis of this definition, a new FTPF is designed:

$$v(t) = \begin{cases} v_0 \cdot \left[ 1 - \tanh\left( \frac{t^n}{T_f - t} \right) \right] + v_{T_f}, & 0 \leq t < T_f \\ v_{T_f}, & t \geq T_f, \end{cases} \tag{6}$$

where $T_f > 0, v_0 > 0, v_{T_f} > 0, 0 < n \leq 1$.

The following is the continuity proof of the proposed FTPF:

For $t < T_f$, we set $f(t) = \tanh(t)$, $g(t) = \frac{t^n}{T_f - t}$, so we yields $v(t) = v_0 \cdot [1 - f(g(t))] + v_{T_f}$ and $g(t) > 0$, $0 < f(g(t)) < 1$.

We can therefore deduce that $v(t) > 0$.

Derivation of (6) gives:

$$\dot{v}(t) = -v_0 f'(g(t)) g'(t), \tag{7}$$

where $g'(t) = \frac{n t^{n-1}(T_f - t) + t^n}{(T_f - t)^2} > 0$, $f'(t) = 1 - \tanh^2(t) > 0$.

For $t < T_f$, it is simple to observe that $v(t) > 0$, $\dot{v}(t) < 0$. For $t \geq T_f$, $v(t) > 0$, $\dot{v}(t) = 0$.

So $v(t)$ satisfies the conditions $(1), (2)$ mentioned in **Definition 2**.

(7) can have the form:

$$\dot{v}(t) = -v_0 \frac{\frac{4n t^{n-1}(T_f - t) + 4t^n}{(t - T_f)^2}}{e^{\frac{2t}{T_f - t}} + e^{\frac{-2t}{T_f - t}} + 2}, \tag{8}$$

in this limit, for $t \to T_f$, $\frac{4t^n}{(t - T_f)^2}$ is a higher order infinity of $\frac{4n t^{n-1}(T_f - t)}{(t - T_f)^2}$.

According to L'Hopital's rule, we can get:

$$\dot{v}(t) = \lim_{t \to T_f^-} -v_0 \frac{\frac{4T_f^n}{(t-T_f)^2}}{e^{\frac{2t}{T_f-t}} + e^{\frac{-2t}{T_f-t}} + 2}$$

$$= \lim_{t \to T_f^-} \frac{\frac{4v_0 T_f^{n-1}}{(t-T_f)}}{e^{\frac{2t}{T_f-t}} - e^{\frac{-2t}{T_f-t}}} \tag{9}$$

$$= \lim_{t \to T_f^-} \frac{-2v_0 T_f^{n-2}}{e^{\frac{2t}{T_f-t}} + e^{\frac{-2t}{T_f-t}}}$$

$$= 0.$$

Therefore, when $t$ approaches $T_f$ from the left

$$\frac{d}{dt}v(t) = 0. \tag{10}$$

Accordingly, $\frac{d}{dt}v(t)$ is continuous.

For derivatives with order greater than 1, and for $t \to T_f^-$, $e^{\frac{-bt}{T_f-t}} \to 0$, $e^{\frac{bt}{T_f-t}} \to +\infty$, where $b > 0$ is a constant.

For $t \to T_f$, the $r$th order derivative of $g(t)$ is:

$$g^{(r)}(t) = \frac{\Upsilon}{(T_f - t)^{r+1}}, \tag{11}$$

where $\Upsilon \in \mathbb{R}$ is an unknown coefficient.

The $r$th order derivative of $f(t)$ can be written as:

$$f^{(r)}(t) = \frac{a(e^t + e^{-t})^c + f(e^t - e^{-t})^c}{(e^t + e^{-t})^{c+2}}, \tag{12}$$

where $a, c, f$ are unknown coefficients.

Finally, we can get the desired result from Bruno's formula [31]:

$$\frac{d^r}{dt^r}f(g(t)) = \sum_{i=1}^{r} f^{(i)}(g(t))B_{r,i}\left(f^{(1)}(t), f^{(2)}(t), \ldots, f^{(r)}(t)\right). \tag{13}$$

Combine the (11) and (12), formula (13) can be rewritten as:

$$\lim_{t \to T_f^-} \frac{d^r}{dt^r} f(g(t)) =$$

$$\lim_{t \to T_f^-} \sum_{i=1}^{r} \frac{\frac{\Upsilon}{(T_f-t)^{i+1}}}{e^{\frac{2t}{T_f-t}}} B_{r,i}\left(f^{(1)}(t), f^{(2)}(t), \ldots, f^{(r)}(t)\right), \tag{14}$$

where $B_{r,i}\left(f^{(1)}(t), f^{(2)}(t), \ldots, f^{(r)}(t)\right)$ is bounded.

Using L'Hopital's rule, we can obtain that for $t \to T_f^-$, $\frac{d^r}{dt^r}f(g(t)) = 0$.

Therefore, $\lim_{t \to T_f^+}\left[\frac{d^r v(t)}{dt^r}\right] = 0$.

And $\lim_{t \to T_f^-}\left[\frac{d^r v(t)}{dt^r}\right] = 0$.

So $v(t)$ is a function that complies with **Definition 2**.

### D. Neural Networks

NNs can approximate unknown nonlinear functions. In this paper, we will employ them to approximate $\xi$:

$$\xi = W^{*T} \Gamma(Z) + \Phi(Z) \quad (|\Phi(Z)| \leq \delta), \tag{15}$$

where $\xi$ denotes the disturbance, $W = [W_1, \cdots, W_p]^T$ is the weight vector of NN and $p$ represents the number of neurons.

$\Gamma(Z) = [\Gamma_1(Z), \cdots, \Gamma_p(Z)]^T$ is the vector of Gaussian basis function as well as $\Phi(Z)$ represents the error from approximation, $\delta$ stands for an accuracy level. The following is the definition of $\Gamma_i(Z)$:

$$\Gamma_i(Z) = \exp\left[-\frac{(Z - C_i)^T (Z - C_i)}{2b_i^2}\right], \tag{16}$$

where $C_i = [C_{i,1}, C_{i,2}, C_{i,3}, \ldots]^T$ is the center of the function and the length of $C_i$ is determined in practice. $b_i$ stands for the function width.

The ideal vector of weight is defined below:

$$W^* = \arg \min_{W \in R^p} \left\{f(Z) - W^T \Gamma(Z)\right\}. \tag{17}$$

Furthermore, the input vector $Z \in \Omega_z$ to be determined in practice.

### III. QUADROTOR UNMANNED AERIAL VEHICLE MODEL

To facilitate the determination of various parameters of QUAV, the position and speed of QUAV are expressed in the earth coordinate system. However, sensor measurements are encoded in the body coordinate system, so a transformation matrix is required to convert these vectors to the earth coordinate system.

**Lemma 3 [32]:** The transformation matrix that converts vectors from the body coordinate system to the earth coordinate system is mathematically defined as follows:

$$R = \begin{bmatrix} C_\theta C_\psi & -C_\phi S_\psi + S_\phi S_\theta C_\psi & S_\phi S_\psi + C_\phi S_\theta C_\psi \\ C_\theta S_\psi & C_\phi C_\psi + S_\phi S_\theta S_\psi & -S_\phi C_\psi + C_\phi S_\theta S_\psi \\ -S_\theta & S_\phi C_\theta & C_\phi C_\theta \end{bmatrix}, \tag{18}$$

where $S_{(x)} = \sin(x), C_{(x)} = \cos(x)$.

To increase the utility of the designed control rate, it is necessary to consider external disturbances. This allows us to obtain the equation for the combined external force and velocity in the QUAV dynamic model, which constitutes the position dynamic model:

$$a = (F/m) \cdot R \cdot e - g \cdot e_3 + (K_F/m) \cdot (v - v_{air}), \tag{19}$$

where $F = c_T \cdot (\omega_1^2 + \omega_2^2 + \omega_3^2 + \omega_4^2)$ is the total lift of the QUAV, $K_F = \text{diag}(K_{Fx}, K_{Fy}, K_{Fz})$ is a coefficient matrix, $e_3 = [0, 0, 1]^T$, $a = [a_x, a_y, a_z]^T$ is the acceleration vector group of QUAV on X-axis, Y-axis and Z-axis. And $c_T$ is the propeller tension coefficient.

The gyroscopic torque, $G_a$, is defined as follows:

$$G_a = \begin{bmatrix} J_t \cdot q \cdot (-\omega_1 + \omega_2 - \omega_3 + \omega_4) \\ J_t \cdot p \cdot (\omega_1 - \omega_2 + \omega_3 - \omega_4) \\ 0 \end{bmatrix}, \tag{20}$$

where $J_t$ stands for the combined moment of inertia of the entire rotor and propeller assembly about the body-frame's rotational axis. Meanwhile, $p$ and $q$ denote the angular velocities corresponding to the roll and pitch angles in the coordinate system of the earth, respectively.

$\varpi$ is the moment generated by the propeller on the QUAV. It can be expressed as follows:

$$\varpi = \begin{bmatrix} d \cdot c_T \cdot \left( -\omega_2^2 + \omega_4^2 \right) \\ d \cdot c_T \cdot \left( -\omega_1^2 + \omega_3^2 \right) \\ c_M \cdot \left( -\omega_1^2 + \omega_2^2 - \omega_3^2 + \omega_4^2 \right) \end{bmatrix}, \quad (21)$$

where $c_M$ is the propeller torque coefficient, $d$ represents the length from the QUAV center to driver.

Combining (20) and (21), the attitude equation of the QUAV can be obtained from Euler's equation:

$$J \cdot \dot{\omega} = G_a + \varpi - \omega \times J \cdot \omega + K_T \cdot (\omega - \omega_{\text{air}}), \quad (22)$$

where $J = diag\left(I_{xx}, I_{yy}, I_{zz}\right)$ is inertia matrix, $\omega$ indicates the angular velocity in the body coordinate system, $K_T = diag\left(K_{T1}, K_{T2}, K_{T3}\right)$ is coefficient matrix.

$\phi$ indicate the roll, $\theta$ represent the pitch and $\psi$ denote the yaw. $r$ represents yaw angle's angular speed in the earth coordinate system. The conversion of $[\phi, \theta, \psi]^T$ to $[p, q, r]^T$ is:

$$\begin{bmatrix} p \\ q \\ r \end{bmatrix} = \begin{bmatrix} 1 & 0 & -S_\theta \\ 0 & C_\phi & S_\phi C_\theta \\ 0 & -S_\phi & C_\phi C_\theta \end{bmatrix} \begin{bmatrix} \dot{\phi} \\ \dot{\theta} \\ \dot{\psi} \end{bmatrix}. \quad (23)$$

And we define:

$$W = \begin{bmatrix} 1 & 0 & -S_\theta \\ 0 & C_\phi & S_\phi C_\theta \\ 0 & -S_\phi & C_\phi C_\theta \end{bmatrix}. \quad (24)$$

Combining (19)-(23), and set $d_1^e = -\frac{K_F}{m} \cdot v_{air}$, $d_2^e = -W \cdot J^{-1} \cdot K_T \cdot \omega_{air}$ to be the external disturbance. The kinetic equation of QUAV can be expressed as:

$$\begin{cases} \ddot{x}^e = u_1^e + \frac{K_F}{m} \cdot \dot{x}^e + d_1^e \\ \ddot{\omega}^e = u_2^e - \zeta \cdot \dot{\omega}^e + d_2^e, \end{cases} \quad (25)$$

where $u_1^e = \left(\frac{F}{m}\right) \cdot R \cdot e_3 - g \cdot e_3$, $u_2^e = W \cdot J^{-1} \cdot \varpi + W \cdot J^{-1} \cdot G_a$, $\zeta = \dot{W}^{-1} \cdot W + W \cdot J^{-1} \cdot M \cdot J \cdot W^{-1} - W \cdot J^{-1} \cdot K_T \cdot W^{-1}$, $x^e$ and $\omega^e$ represent the position state and attitude state of the QUAV in the earth coordinate system, respectively.

M is a skew-symmetric matrix:

$$M = \begin{bmatrix} 0 & -r & q \\ r & 0 & -p \\ -q & p & 0 \end{bmatrix}. \quad (26)$$

Therefore, the equation of state of the QUAV can be deduced as:

$$\begin{bmatrix} \ddot{x}^e \\ \ddot{\omega}^e \end{bmatrix} = \begin{bmatrix} u_1^e \\ u_2^e \end{bmatrix} + \begin{bmatrix} \frac{K_F}{m} & 0 \\ 0 & -\zeta \end{bmatrix} \begin{bmatrix} \dot{x}^e \\ \dot{\omega}^e \end{bmatrix} + \begin{bmatrix} d_1^e \\ d_2^e \end{bmatrix}. \quad (27)$$

So, the $i$th QUAV's state characteristic of the system is as follows:

$$\begin{cases} \dot{x}_{i,0} = x_{i,1} \\ \dot{x}_{i,1} = u_i + s_i + \xi_i, \end{cases} \quad (28)$$

where $\dot{x}_{i,0}$, $\dot{x}_{i,1}$, $u_i$ and $s_i$ are the vector of position, vector of velocity, vector of input and matrix describing the model of the $i$th QUAV, respectively.

## IV. Control Law of Quadrotor Unmanned Aerial Vehicle Swarms

Setting the global reference signal:

$$\begin{cases} \dot{x}_{0,0} = x_{0,1} \\ \dot{x}_{0,1} = x_{0,2}, \end{cases} \quad (29)$$

where $x_{0,0}$, $x_{0,1}$ and $x_{0,2}$ are global position reference signal, global velocity reference signal and global acceleration reference signal, respectively.

Since the QUAV is considered as an underdriven system, the $\phi$ and $\theta$ need to be determined based on the desired position tracking. The required attitude signals are generated using a filtered inversion technique. The derivative signals of the attitude command are obtained using a linear differential tracker.

QUAV formation's error is defined below:

$$\begin{cases} e_{i,0}(t) = x_{i,0} - x_{0,0} - h_{i,0} \\ e_{i,1}(t) = x_{i,1} - x_{0,1} - h_{i,1}, \end{cases} \quad (30)$$

where $h_{i,j}(j = 1, 2)$ are the formation functions.

Thus we can get the $\tau$th order error of the $i$th QUAV as:

$$\begin{aligned} E_{i,\tau} &= \sum_{j=1}^{N} \bar{a}_{ij} \left( x_{i,\tau} - h_{i,\tau} - x_{j,\tau} + h_{j,\tau} \right) \\ &\quad + b_i \left( x_{i,\tau} - x_{0,\tau} - h_{i,\tau} \right) \\ &= \sum_{j=1}^{N} \bar{a}_{ij} \left( x_{i,\tau} - x_{j,\tau} \right) + b_i \left( x_{i,\tau} - x_{0,\tau} \right) + f_{i,\tau}, \end{aligned} \quad (31)$$

where $f_{i,\tau} = \sum_{j=1}^{N} \bar{a}_{ij} \left( -h_{i,\tau} + h_{j,\tau} \right) + b_i \left( -h_{i,\tau} \right)$.

The error is as follows:

$$E_\tau = H \otimes I_N \cdot (x_\tau - 1_N \otimes x_{0,\tau}) + f_\tau, \quad (32)$$

where $H = L + B$.

### A. Error Transfer

The error transfer function $T(\varepsilon)$ is used to convert the error for each QUAV:

$$T(\varepsilon) = \frac{e^\varepsilon - e^{-\varepsilon}}{e^\varepsilon + e^{-\varepsilon}}. \quad (33)$$

Therefore, the transfer error $\varepsilon_{i,k,0}$ for the $k$th degree of freedom of the $i$th QUAV could be derived from the equation below:

$$E_{i,k,0}(t) = v(t) \cdot T(\varepsilon_{i,k,0}). \quad (34)$$

Combine (33) and (34), we can obtain the transfer error $\varepsilon_{i,k,0}$ expression:

$$\varepsilon_{i,k,0} = \frac{1}{2} \ln \left( \frac{v(t) + E_{i,k,0}(t)}{v(t) - E_{i,k,0}(t)} \right). \quad (35)$$

The derivative of the transfer error $\varepsilon_{i,k,1}$ can be obtained by simultaneously deriving both sides of (34):

$$\begin{aligned} \dot{\varepsilon}_{i,k,0} = \varepsilon_{i,k,1} &= -\frac{\dot{v}(t) \cdot T\left(\varepsilon_{i,k,0}\right)}{v(t) \cdot \frac{\partial T(\varepsilon_{i,k,0})}{\partial \varepsilon_{i,k,0}}} + \frac{E_{i,k,1}(t)}{v(t) \cdot \frac{\partial T(\varepsilon_{i,k,0})}{\partial \varepsilon_{i,k,0}}} \\ &= \gamma_{ik} + F_{ik} \cdot E_{i,k,1}(t), \end{aligned} \quad (36)$$

where $\gamma_{ik} = -\frac{\dot{v}(t) \cdot T(\varepsilon_{i,k,0})}{v(t) \cdot \frac{\partial T(\varepsilon_{i,k,0})}{\partial \varepsilon_{i,k,0}}}$, $F_{ik} = \frac{1}{v(t) \cdot \frac{\partial T(\varepsilon_{i,k,0})}{\partial \varepsilon_{i,k,0}}}$.

**Remark 2:** $v(t) > 0$, $\frac{\partial T(\varepsilon_0)}{\partial \varepsilon_0} = \frac{4}{(e^\varepsilon + e^{-\varepsilon})^2} > 0$. Therefore $F > 0$.

Therefore, form (36), the first order derivative of formation transfer error with finite-time prescribed performance can be characterized as:

$$\varepsilon_1 = \gamma + F\left[H \otimes I_N \cdot (x_1 - 1_N \otimes x_{0,1}) + f_1\right], \quad (37)$$

where

$$\begin{cases} \gamma_i = [\gamma_{i1}, \gamma_{i2}, \gamma_{i3}, \gamma_{i4}, \gamma_{i5}, \gamma_{i6}] \\ \gamma = [\gamma_1, \gamma_2, \cdots \gamma_N]^T \\ F_i = \text{diag}\left([F_{i1}, F_{i2}, F_{i3}, F_{i4}, F_{i5}, F_{i6}]\right) \\ F = \text{diag}\left([F_1, F_2, F_3, \cdots, F_N]\right). \end{cases} \quad (38)$$

### B. Design of Control Law

We set the sliding error $S$ as:

$$S = \beta \varepsilon_0 + \varepsilon_1, \quad (39)$$

where

$$\begin{cases} \beta = \text{diag}\left([\beta_1, \beta_2, \cdots, \beta_N]\right) \\ \beta_i = \text{diag}\left([\beta_{i1}, \beta_{i2}, \cdots, \beta_{i6}]\right) \\ S = \left[S_1^T, S_2^T, \cdots, S_N^T\right]^T \\ S_i = [S_{i1}, S_{i2}, \cdots, S_{i6}]^T. \end{cases} \quad (40)$$

The chosen Lyapunov function $V_{eq}$ is as follows:

$$V_{eq} = \frac{1}{2} S^T S. \quad (41)$$

Setting $u^{eq}$ to represent the equivalent input, and $u^s$ is designed as the switch input, and the control law is:

$$u = u^{eq} + u^s. \quad (42)$$

Derivation of equation (41) yields:

$$\begin{aligned} \dot{V}_{eq} = S^T \{ &\beta \varepsilon_1 + \dot{\gamma} + \dot{F} E_1 \\ &+ F\left[H \otimes I_N \cdot (u + s + \xi - 1_N \otimes x_{0,2}) + f_2\right] \}. \end{aligned} \quad (43)$$

Then the control law $u^{eq}$ is:

$$\begin{aligned} u^{eq} = &1_N \otimes x_{0,2} - s - H^{-1} \otimes I_N \cdot f_2 \\ &- F^{-1} \cdot H^{-1} \otimes I_N \cdot \left(\beta \varepsilon_1 + \dot{\gamma} + \dot{F} E_1\right). \end{aligned} \quad (44)$$

Then, (43) can be rewritten as follows:

$$\dot{V}_{eq} = S^T F \cdot H \otimes I_N \cdot (u^s + \xi). \quad (45)$$

(45) can be written as:

$$\dot{V}_{eq} = \sum_{i=1}^{N} S_i^T F_i \left(u_i^s + \xi_i - \sum_{j=1}^{N} \bar{a}_{ij} \left(u_j^s + \xi_j\right)\right). \quad (46)$$

For formation control, it is essential for the followers to track the leader. Therefore, the control law for the leader needs to be solved first.

According to (46), we can rewrite the leader's Lyapunov function as follows:

$$\dot{V}_{le} = S_{le}^T F_{le} \left(u_{le}^s + \xi_{le}\right). \quad (47)$$

Since NNs are used to approximate the nonlinear external disturbance:

$$\xi_{le,i} = W_{le,i}^{*}{}^T \Gamma_{le,i}\left(Z_{le,i}\right) + \Phi\left(Z_{le,i}\right)). \quad (48)$$

where $|\Phi\left(Z_{le,i}\right)| \leq \delta_{le,i}$ and $\xi_{le}$ is defined as: $\xi_{le} = [\xi_{le,1}, \xi_{le,2}, \xi_{le,3}, \xi_{le,4}, \xi_{le,5}, \xi_{le,6}]^T$.

With added NNs, the Lyapunov function is updated as:

$$\dot{V}_{le} = S_{le}^T F_{le} \left[u_{le}^s + \xi_{le}\right] + \sum_{k=1}^{6} \left(-\tilde{\eta}_{le,k} \dot{\hat{\eta}}_{le,k}\right). \quad (49)$$

where $\hat{\eta}_{le,i} = \left\|\hat{W}_{le,i}^T\right\|^2$, $\tilde{\eta}_{le,i} = \eta_{le,i} - \hat{\eta}_{le,i}$.

$$\begin{aligned} \dot{V}_{le} = &\sum_{k=1}^{6} S_{le,k} F_{le,k} [u_{le,k}^s + W_{le,k}^{*}{}^T \Gamma_{le,k}\left(Z_{le,k}\right) \\ &+ \Phi\left(Z_{le,k}\right)] + \sum_{k=1}^{6} \left(-\tilde{\eta}_{le,k} \dot{\hat{\eta}}_{le,k}\right). \end{aligned} \quad (50)$$

$$\begin{aligned} \dot{V}_{le} = &\sum_{k=1}^{6} \{ S_{le,k} F_{le,k} [u_{le,k}^s + W_{le,k}^{*}{}^T \Gamma_{le,k}\left(Z_{le,k}\right) \\ &+ \Phi\left(Z_{le,k}\right)] - \tilde{\eta}_{le,k} \dot{\hat{\eta}}_{le,k} \}. \end{aligned} \quad (51)$$

From **Lemma 2**, we can deduce that:

$$\begin{aligned} &S_{le,k} F_{le,k} W_{le,k}^{*}{}^T \Gamma_{le,k}\left(Z_{le,k}\right) \\ &\leq [S_{le,k}^2 F_{le,k}^2 \|W_{le,k}^*\|^2 \\ &\quad \Gamma_{le,k}^T\left(Z_{le,k}\right) \Gamma_{le,k}\left(Z_{le,k}\right)]/4\varsigma_1 + \varsigma_1 \\ &\leq [S_{le,k}^2 F_{le,k}^2 \eta_{le,i} \\ &\quad \Gamma_{le,k}^T(Z_{le,k}) \Gamma_{le,k}\left(Z_{le,k}\right)]/4\varsigma_1 + \varsigma_1. \end{aligned} \quad (52)$$

$$S_{le,k} F_{le,k} \Phi\left(Z_{le,k}\right) \leq \frac{S_{le,k}^3}{3} + \frac{F_{le,k}^3}{3} + \frac{\delta_{le,i}^3}{3}. \quad (53)$$

Substituting (53), (52) into (51), we get:

$$\begin{aligned} \dot{V}_{le} \leq &\sum_{k=1}^{6} \{ S_{le,k} F_{le,k} u_{le,k}^s \\ &+ \left[S_{le,k}^2 F_{le,k}^2 \eta_{le,i} \Gamma_{le,k}^T\left(Z_{le,k}\right) \Gamma_{le,k}\left(Z_{le,k}\right)\right]/4\varsigma_1 \\ &+ \varsigma_1 + \frac{S_{le,k}^3}{3} + \frac{F_{le,k}^3}{3} + \frac{\delta_{le,i}^3}{3} - \tilde{\eta}_{le,k} \dot{\hat{\eta}}_{le,k} \}. \end{aligned} \quad (54)$$

Thus the control rate $u_{le,k}^s$ can be given as:

$$\begin{aligned} u_{le,k}^s = &- [S_{le,k}^2 F_{le,k}^2 \eta_{le,i} \\ &\quad \Gamma_{le,k}^T\left(Z_{le,k}\right) \Gamma_{le,k}\left(Z_{le,k}\right)]/4\varsigma_1 \\ &- \frac{S_{le,k}^2}{3 F_{le,k}} - \mu_{le,k} S_{le,k}, \end{aligned} \quad (55)$$

where $Z_{le,k} = [x_{0,0,k}, x_{0,1,k}, x_{0,2,k}, x_{le,0,k}, x_{le,1,k}, S_{le,k}]^T$, $\mu_{le,k} > 0$.

The corresponding parameter update rate of the NNs can be:

$$\begin{aligned} \dot{\hat{\eta}}_{le,k} = &\left[S_{le,k}^2 F_{le,k}^2 \Gamma_{le,k}^T\left(Z_{le,k}\right) \Gamma_{le,k}\left(Z_{le,k}\right)\right]/4\varsigma_1 \\ &- \omega_{le,k} \hat{\eta}_{le,k}. \end{aligned} \quad (56)$$

where $\omega_{le,k} > 0$.

Let the leader's $u^s$ be:

$$u_{le}^s = \left[u_{le,1}^s, u_{le,2}^s, u_{le,3}^s, u_{le,4}^s, u_{le,5}^s, u_{le,6}^s\right]^T. \quad (57)$$

Substituting (55), (56) into (54) yields:

$$
\begin{aligned}
\dot{V}_{le} \leq &\sum_{k=1}^{6} \left(-\mu_{le,k} F_{le,k} S_{le,k}^2\right) \\
&+ \sum_{k=1}^{6} \left(-\omega_{le,k} \tilde{\eta}_{le,k}^2\right) + \sum_{k=1}^{6} \Delta_{le,k},
\end{aligned}
\quad (58)
$$

where $\Delta_{le,k} = \varsigma_1 + \frac{F_{le,k}^3}{3} + \frac{\delta_{le,i}^3}{3} + \omega_{le,k} \tilde{\eta}_{le,k} \eta_{le,k}$.

The follower's Lyapunov function can be rewritten as:

$$
\begin{aligned}
\dot{V}_f = &\sum_{i=1}^{N-1} S_{f,i}^T F_{f,i} \left[u_{f,i}^s + \xi_{f,i} - \sum_{j=1}^{N} \bar{a}_{ij} \left(u_j^s + \xi_j\right)\right] \\
&+ \sum_{k=1}^{6} \left(-\tilde{\eta}_{f,i,k} \dot{\hat{\eta}}_{f,i,k}\right).
\end{aligned}
\quad (59)
$$

Noting the complexity of the form, the control rates were designed as follows:

$$u_{f,i}^s = u_{f,i,1}^s + u_{f,i,2}^s, \quad (60)$$

where

$$u_{f,i,1}^s = \sum_{j=1}^{N} \bar{a}_{ij} u_j^s. \quad (61)$$

$u_{f,i,1}^s$ is related to the control rate of the leader and other followers, so the control rate of different intelligences should be designed according to the graph theory successively in actual control.

Substituting (60), (61) into (59) gives:

$$
\begin{aligned}
\dot{V}_f = &\sum_{i=1}^{N-1} [S_{f,i}^T F_{f,i} (u_{f,i,2}^s + \xi_{f,i} - \sum_{j=1}^{N} \bar{a}_{ij} \xi_j) \\
&+ \sum_{k=1}^{6} (-\tilde{\eta}_{f,i,k} \dot{\hat{\eta}}_{f,i,k})].
\end{aligned}
\quad (62)
$$

Let $Q_{f,i} = -\sum_{j=1}^{N} \bar{a}_{ij} \xi_j$, $Q \in R^{6 \times 1}$. We can deduce that $|Q_{f,i,k}| \leq -\sum_{j=1}^{N} \bar{a}_{ij} \|\xi_j\| \leq q_{f,i,k}$.

$\xi_{f,i,k}$ is the external disturbance of the $i$th follower, again using a NN to approximate its value.

$$\xi_{f,i,k} = W_{f,i,k}^{*}{}^T \Gamma_{f,i,k} \left(Z_{f,i,k}\right) + \Phi\left(Z_{f,i,k}\right), \quad (63)$$

where $|\Phi\left(Z_{f,i,k}\right)| \leq \delta_{f,i,k}$, $Z_{f,i,k} = [x_{0,0,k} + h_{i,0,k}, x_{0,1,k} + h_{i,1,k}, x_{0,2,k} + h_{i,2,k}, x_{f,i,1,k}, x_{f,i,1,k}, S_{f,i,k}]^T$.

Substitute (63) into (62):

$$
\begin{aligned}
\dot{V}_f = &\sum_{i=1}^{N-1} \sum_{k=1}^{6} \{S_{f,i,k} F_{f,i,k} [u_{f,i,2,k}^s \\
&+ W_{f,i,k}^{*}{}^T \Gamma_{f,i,k}(Z_{f,i,k}) + \Phi\left(Z_{f,i,k}\right) + Q_{f,i,k}] \\
&- \tilde{\eta}_{f,i,k} \dot{\hat{\eta}}_{f,i,k}\}.
\end{aligned}
\quad (64)
$$

Based on **Lemma 2**, we can deduce that:

$$
\begin{aligned}
S_{f,i,k} &F_{f,i,k} W_{f,i,k}^{*}{}^T \Gamma_{f,i,k} \left(Z_{f,i,k}\right) \\
&\leq [S_{f,i,k}^2 F_{f,i,k}^2 \eta_{f,i,k} \\
&\quad \Gamma_{f,i,k}^T(Z_{f,i,k}) \Gamma_{f,i,k} \left(Z_{f,i,k}\right)] / 4\varsigma_2 + \varsigma_2.
\end{aligned}
\quad (65)
$$

$$S_{f,i,k} F_{f,i,k} Q_{f,i,k} \leq \frac{S_{f,i,k}^3}{3} + \frac{F_{f,i,k}^3}{3} + \frac{q_{f,i,k}^3}{3}. \quad (66)$$

$$S_{f,i,k} F_{f,i,k} \Phi\left(Z_{f,i,k}\right) \leq \frac{S_{f,i,k}^3}{3} + \frac{F_{f,i,k}^3}{3} + \frac{\delta_{f,i,k}^3}{3}. \quad (67)$$

Substituting (65), (66), and (67) into (64) yields:

$$
\begin{aligned}
\dot{V}_f \leq &\sum_{i=1}^{N-1} \sum_{k=1}^{6} \{S_{f,i,k} F_{f,i,k} u_{f,i,2,k}^s + \frac{2S_{f,i,k}^3}{3} \\
&+ \frac{2F_{f,i,k}^3}{3} + \frac{\delta_{f,i,k}^3}{3} + \frac{q_{f,i,k}^3}{3} - \tilde{\eta}_{f,i,k} \dot{\hat{\eta}}_{f,i,k} \\
&+ [S_{f,i,k}^2 F_{f,i,k}^2 \eta_{f,i,k} \\
&\quad \Gamma_{f,i,k}^T \left(Z_{f,i,k}\right) \Gamma_{f,i,k}(Z_{f,i,k})] / 4\varsigma_2 + \varsigma_2\}.
\end{aligned}
\quad (68)
$$

Thus, the control rate of the $i$th follower is:

$$
\begin{aligned}
u_{f,i,2,k}^s = &- [S_{f,i,k} F_{f,i,k} \eta_{f,i,k} \\
&\quad \Gamma_{f,i,k}^T \left(Z_{f,i,k}\right) \Gamma_{f,i,k}(Z_{f,i,k})] / 4\varsigma_2 \\
&- \frac{2S_{f,i,k}^2}{3F_{f,i,k}} - \mu_{f,i,k} S_{f,i,k}.
\end{aligned}
\quad (69)
$$

The relative NNs parameter update rates are:

$$
\begin{aligned}
\dot{\hat{\eta}}_{f,i,k} = &[S_{f,i,k}^2 F_{f,i,k}^2 \\
&\quad \Gamma_{f,i,k}^T \left(Z_{f,i,k}\right) \Gamma_{f,i,k} \left(Z_{f,i,k}\right)] / 4\varsigma_2 \\
&- \omega_{f,i,k} \hat{\eta}_{f,i,k},
\end{aligned}
\quad (70)
$$

where $\mu_{f,i,k} > 0$, $\quad \omega_{f,i,k} > 0$.

Substituting (69), (70) into (68) yields:

$$
\begin{aligned}
\dot{V}_f \leq &\sum_{i=1}^{N-1} \sum_{k=1}^{6} \left(-\mu_{f,i,k} F_{f,i,k} S_{f,i,k}^2\right) \\
&+ \sum_{i=1}^{N-1} \sum_{k=1}^{6} \left(-\omega_{f,i,k} \tilde{\eta}_{f,i,k}^2\right) + \sum_{i=1}^{N-1} \sum_{k=1}^{6} \left(\Delta_{f,i,k}\right),
\end{aligned}
\quad (71)
$$

where $\Delta_{f,i,k} = \varsigma_2 + \frac{2F_{f,i,k}^3}{3} + \frac{\delta_{f,i,k}^3}{3} + \frac{q_{f,i,k}^3}{3} + \omega_{f,i,k} \tilde{\eta}_{f,i,k} \eta_{f,i,k}$.

The $i$th follower's $u_{f,i}^s$ can be expressed as follows:

$$u_{f,i}^s = u_{f,i,1}^s + u_{f,i,2}^s, \quad (72)$$

where $u_{f,i,k}^s = \left[u_{f,i,k,1}^s, u_{f,i,k,2}^s, \dots, u_{f,i,k,6}^s\right]^T$.

After obtaining the control law $u_{le}^s$ for the leader, the overall control input $u^s$ for all QUAVs can be obtained by combining the input of leader and the followers' control inputs, denoted as $u_{f,i}^s$ in a specific order according to graph theory.

Therefore, we can get all the overall control rates based on (42).

The seleted Lyapunov function is

$$V = \frac{1}{2}S^T S + \frac{1}{2}\sum_{i=1}^{6N}\tilde{\eta}_i^2. \tag{73}$$

Based on (58) and (71), we obtain:

$$\begin{aligned}
\dot{V} \leq &\sum_{k=1}^{6}\left(-\mu_{le,k}F_{le,k}S_{le,k}^2\right) + \sum_{k=1}^{6}\left(-\omega_{le,k}\tilde{\eta}_{le,k}^2\right) \\
&+ \sum_{i=1}^{N-1}\sum_{k=1}^{6}\left(-\mu_{f,i,k}F_{f,i,k}S_{f,i,k}^2\right) \\
&+ \sum_{i=1}^{N-1}\sum_{k=1}^{6}\left(-\omega_{f,i,k}\tilde{\eta}_{f,i,k}^2\right) + \sum_{k=1}^{6}\Delta_{le,k} \\
&+ \sum_{i=1}^{N-1}\sum_{k=1}^{6}\Delta_{f,i,k}.
\end{aligned} \tag{74}$$

So, we can deduce that:

$$\begin{aligned}
\dot{V} \leq &-\mu F\sum_{k=1}^{6N}S_k^2 - \omega\sum_{k=1}^{6N}\tilde{\eta}_k^2 + \sum_{k=1}^{6}\Delta_{le,k} \\
&+ \sum_{i=1}^{N-1}\sum_{k=1}^{6}\left(\Delta_{f,i,k}\right),
\end{aligned} \tag{75}$$

where

$$\begin{cases}
\mu = \min\left(\mu_{le,1}, \mu_{le,2}, \cdots, \mu_{le,6}, \mu_{f,1,1}, \mu_{f,1,2}, \cdots\right) \\
F = \min\left(F_{le,1}, F_{le,2}, \cdots, F_{le,6}, F_{f,1,1}, F_{f,1,2}, \cdots\right) \\
\omega = \min\left(\omega_{le,1}, \omega_{le,2}, \cdots, \omega_{le,6}, \omega_{f,1,1}, \omega_{f,1,2}, \cdots\right).
\end{cases} \tag{76}$$

$$\begin{aligned}
\dot{V} \leq &-\mu F S^T S - \omega\sum_{k=1}^{6N}\tilde{\eta}_k^2 \\
&+ \sum_{k=1}^{6}\Delta_{le,k} + \sum_{i=1}^{N-1}\sum_{k=1}^{6}\left(\Delta_{f,i,k}\right) \\
\leq &-\lambda V + \Delta,
\end{aligned} \tag{77}$$

where $\lambda = \min(2\mu F, 2\omega) > 0$, $\Delta = \sum_{k=1}^{6}\Delta_{le,k} + \sum_{i=1}^{N-1}\sum_{k=1}^{6}\Delta_{f,i,k}$.

**Remark 3:** The advantages of the Lyapunov function used in this paper over existing finite-time control results (see **Lemma 1**) are reflected in the fact that it is more direct, less restricted, and avoids ambiguity issues.

Utilizing the above theory, the errors $S$ and $\tilde{\eta}_i$ are guaranteed to converge in finite time.

## V. EXAMPLE OF SIMULATION

This section validates the feasibility of the designed control strategy through simulation. The QUAVs system realizes the desired effect according to the formation function during the tracking process.

Considering the system has six QUAVs, the initial position of the six QUAVs are as follows:

$$\begin{aligned}
x_{1,0} &= [0.97, 1.78, 0.46, 0.57, 0.65, 0.58]^T; \\
x_{2,0} &= [0.15, 0.92, 1.59, 0.55, 0.35, 0.49]^T; \\
x_{3,0} &= [0.34, 1.78, 0, 0.43, 0.92, 0.55]^T; \\
x_{4,0} &= [1.71, 0.06, 0, 0.77, 0.46, -0.67]^T; \\
x_{5,0} &= [1.23, 1.95, 0.17, 0.50, 0.34, 0.38]^T; \\
x_{6,0} &= [1.81, 1.65, 1.95, 0.86, 0.89, 0.46]^T;
\end{aligned} \tag{78}$$

The initial velocity of the six QUAVs would be 0.

$x_{0,0} = [cos(0.1\pi t), sin(0.1\pi t), sin(0.1\pi t)]^T$ is the global position reference signal of the QUAV. $\psi = 0.2cos(0.1\pi t)$ is the global yaw reference signal. As for the required values for roll and pitch, we will get them using the methods mentioned above.

To enable formation control of QUAVs time-vary formation function, we design the time-vary formation function as follows:

$$\begin{aligned}
h_{1,0} =&[cos(0.3t) - 1, 2sin(0.4t) - 1, cos(0.2t) - 2, \\
&0, 0, 0]^T, \\
h_{2,0} =&[-cos(0.3t) + 1, -2sin(0.2t) + 1, -cos(0.2t) + 2, \\
&0, 0, 0]^T, \\
h_{3,0} =&[sin(0.4t) - 1, cos(0.3t) + 1, -cos(0.2t), \\
&0, 0, 0]^T, \\
h_{4,0} =&[sin(0.3t) - 4, 2cos(0.4t) + 2, sin(0.2t) + 1, \\
&0, 0, 0]^T, \\
h_{5,0} =&[-sin(0.3t) + 4, -2cos(0.4t) - 2, -sin(0.2t) - 1, \\
&0, 0, 0]^T, \\
h_{6,0} =&[0, 0, 0, 0, 0, 0]^T.
\end{aligned} \tag{79}$$

Let $C_i = [-2, -1.5, -1, -0.5, 0, 0.5, 1, 1.5, 2]$ and $b_i = 1$, where $i = 1, 2, \cdots, 6$. Let updating factor $\mu_{le,k} = \mu_{f,i,k} = 25$, $\omega_{le,k} = \omega_{f,i,k} = 0.25$. Let $\varsigma_1 = \varsigma_2 = 0.025$.

For QUAVs' modeling, $m = 2$, $g = 9.8$, $J_t = 1$, $d = 0.2$, $c_T = 2$, $c_M = 1.5$ $K_F = \text{diag}([0.02, 0.02, 0.02])$, $K_T = \text{diag}([0.015, 0.015, 0.015])$, $J = \text{diag}([1.5, 1.5, 2.2])$.

For the parameters in the $v(t)$, it is designed as $n = 0.75$, $v_0 = 10$, $v_{Tf} = 0.2$, and $T_f$ will be prescribed later.

In order for the system to reach convergence in a prescribed time, we design the coefficients of the SMC errors as $\beta_i = [\frac{2}{T_f}; \frac{2}{T_f}; \frac{2}{T_f}; \frac{2}{T_f}; \frac{2}{T_f}; \frac{2}{T_f}]^T$.

With using the 6th QUAV as a leader, the communication matrix is:

$$A = \begin{pmatrix}
0 & 0 & 0 & 0 & 0 & 1 \\
0 & 0 & 0 & 0 & 0.5 & 0.5 \\
0 & 0 & 0 & 0.5 & 0 & 0.5 \\
0.5 & 0 & 0 & 0 & 0 & 0.5 \\
0.5 & 0 & 0 & 0 & 0 & 0.5 \\
0 & 0 & 0 & 0 & 0 & 0
\end{pmatrix} \tag{80}$$

The primary findings are depicted in Fig.1-Fig.11. In Fig.1, QUAVs' position states along the X-axis during formation control with nonlinear disturbances are illustrated. Fig.2 portrays the position errors along the X-axis, demonstrating that

the QUAVs consistently maintain their position within the specified range. Fig.3 and Fig.4 present the position states and errors of QUAVs along the Y-axis, respectively, with Fig.4 indicating the QUAV's errors to converge within the prescribed region. Fig.5 and Fig.6 showcase the position states and errors along the Z-axis, respectively, highlighting the system's convergence within the specified region. Fig.7 and Fig.8 exhibit the yaw angle status and yaw angle errors of QUAVs, with Fig.8 indicating the QUAVs' adherence to the prescribed trajectory under the suggested control strategy. Finally, Fig.9 displays the control inputs in the X-axis direction, revealing that the system maintains acceptable input performance even when subjected to unknown nonlinear function disturbances.

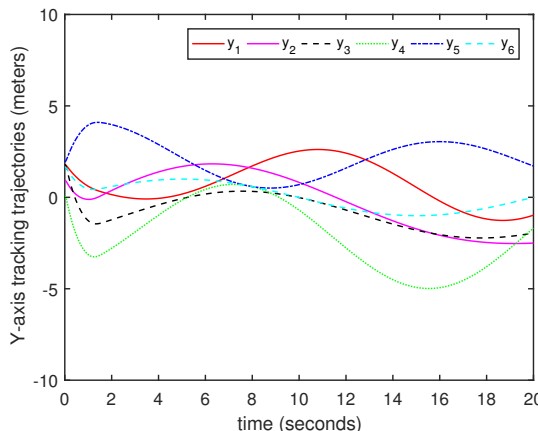

Fig. 3. The position states of the QUAVs along the Y-axis. $y_i$ represents the positional state of the $i$th QUAV.

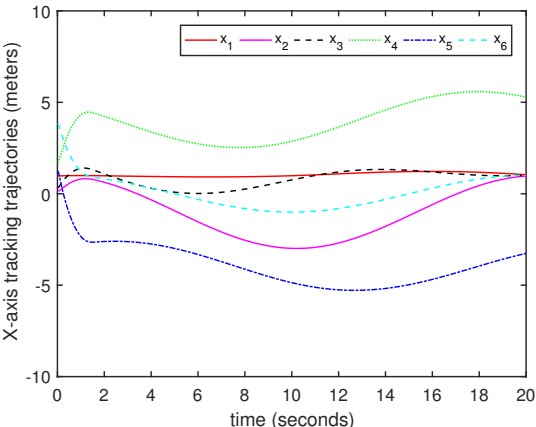

Fig. 1. The position states of the QUAVs along the X-axis. $x_i$ represents the positional state of the $i$th QUAV.

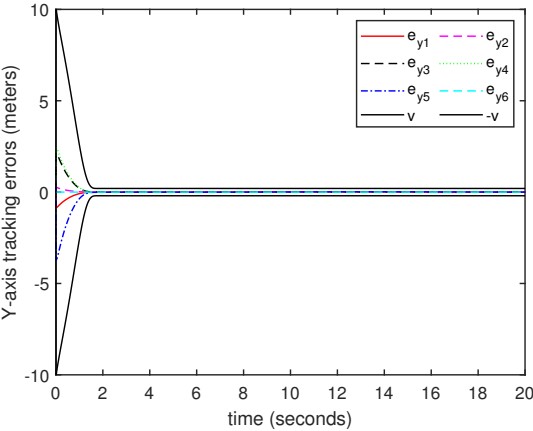

Fig. 4. The position errors of the QUAVs along the Y-axis. $e_{yi}$ represents the positional error of the $i$th QUAV.

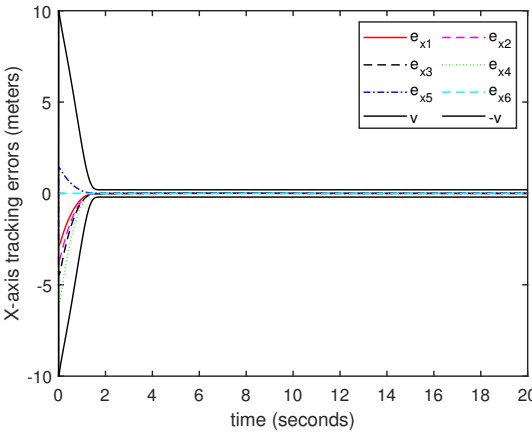

Fig. 2. The position errors of the QUAVs along the X-axis. $e_{xi}$ represents the positional error of the $i$th QUAV.

The simulation result is presented in Fig.10, it reveals the $4th$ QUAV's position tracking error on the Y-axis in the case of $T_f = 2$, $T_f = 3$, $T_f = 5$, and $T_f = 8$, illustrating that the convergence speed of the QUAV varies with the designated time, further emphasizing that the FTPF proposed is capable of effectively controlling convergence speed.

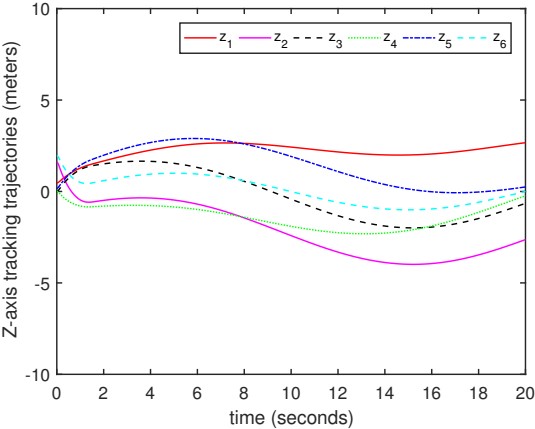

Fig. 5. The positional state of each QUAV along the Z-axis. $z_i$ represents the positional state of the $i$th QUAV.

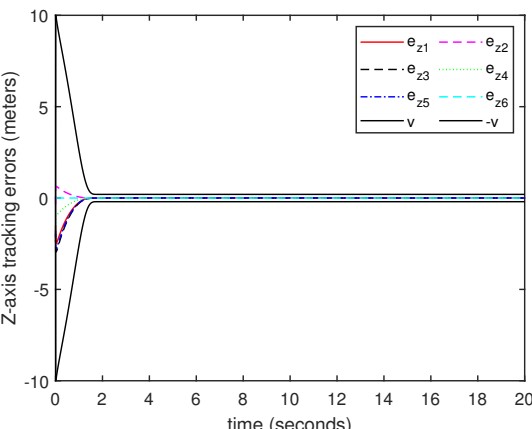

Fig. 6. The position errors of the QUAVs along the Z-axis. $e_{zi}$ represents the positional error of the $i$th QUAV.

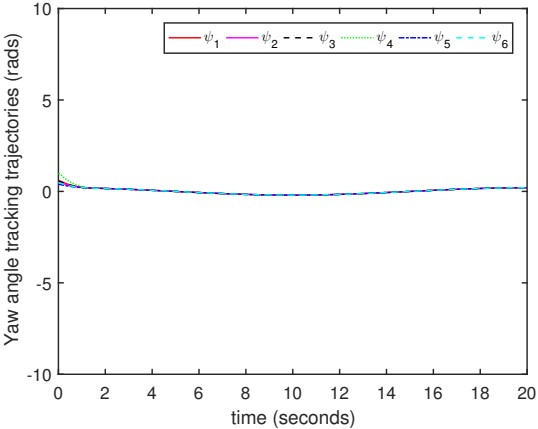

Fig. 7. The attitude states of QUAVs along the yaw angle. $\psi_i$ represents the angular state of the $i$th QUAV.

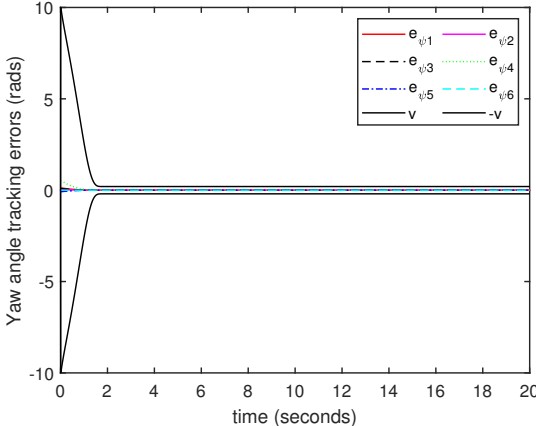

Fig. 8. The attitude errors of the QUAVs along the yaw angle. $e_{\psi i}$ represents the angular error of the $i$th QUAV.

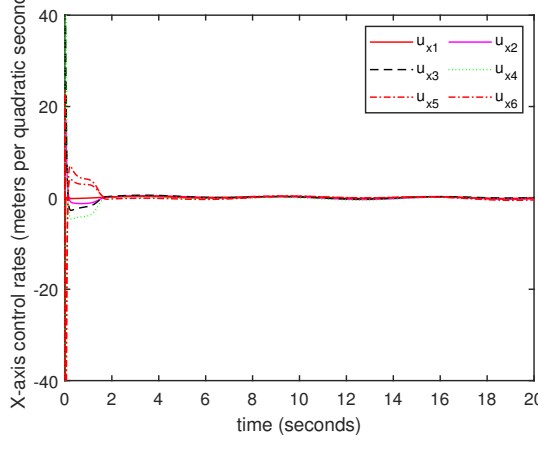

Fig. 9. The control input information of the QUAVs along the X-axis. $u_{x1}$ represents the input of the $i$th QUAV.

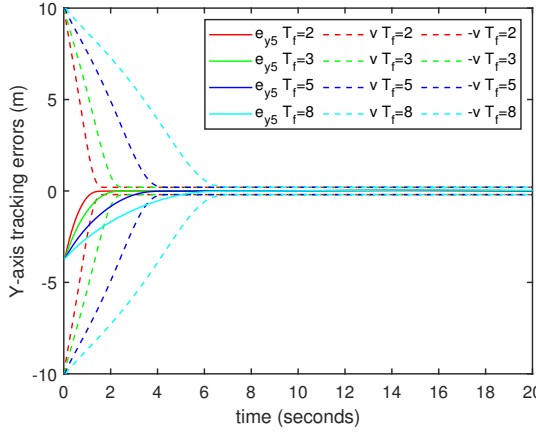

Fig. 10. The position tracking error of the $4th$ QUAV along the Y-axis using proposed FTPF with $T_f = 2$, $T_f = 3$, $T_f = 5$, and $T_f = 8$, respectively.

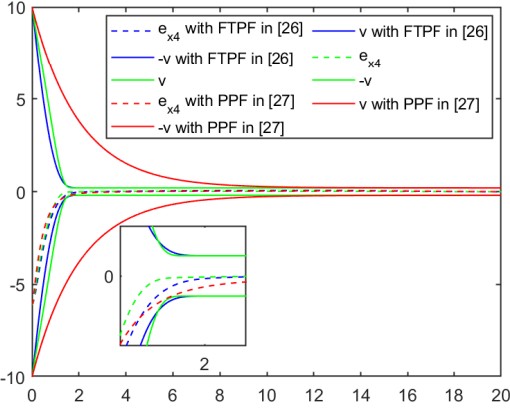

Fig. 11. The position tracking errors of the $4th$ QUAV along the X-axis with the proposed FTPF, the other FTPF in [?] and the prescribed performance function in [?].

Fig.11 presents the simulation results of the position tracking error along the X-axis for the $4th$ agent using the FTPF designed in this paper, the other FTPF in [26], and the prescribed performance function in [27] when $T_f = 2$. From the figures, it can be found that the performance of QUAVs is acceptable after the proposed control law is employed, despite the presence of uncertain nonlinear disturbances. Analysis of Fig.11 reveals that the FTPF proposed in this study offers superior stability in comparison to other finite-time prescribed performance functions. Unlike the slight fluctuations observed in the error of the other FTPF, the FTPF maintains a more consistent trajectory. Additionally, in contrast to the PPF, the FTPF achieves quicker convergence with reduced error. Thus, the FTPF introduced in this work effectively manages convergence rate of the system and maintains error within a narrow region.

## VI. CONCLUSION

This paper focuses on the time-varying control of QUAVs system with FTPF. NNs are used to approximate the disturbances to improve the control accuracy and eliminate chattering problem. Combining the proposed new FTPF and sliding mode control method, a finite-time prescribed performance time-varying formation strategy is proposed. The convergence of the controller is proved by a strict theoretical derivation on the basis of the Lyapunov stability criterion. Finally, simulations are conducted to validate the practicality of the designed control scheme in achieving prescribed performance within a specified time range while maintaining acceptable input performance, even with unknown nonlinear disturbances.

## ACKNOWLEDGMENT

Dianbiao Dong is supported by the National Natural Science Foundation of China [Grant No. 62203356].

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
