# OpenReview forum: "Neural networks adaptive time-varying formation control for quadrotor unmanned aerial vehicles with finite-time prescribed performance"
_IEEE.org/ICIST/2024/Conference — IEEE ICIST 2024 Conference Submission_

### Official Review · Reviewer_fHgM · 2024-08-22
**The topic of the paper is of good interest for the readers.**

**Rating:** 7
**Confidence:** 3

**Review:**

1.The engineering background of the proposed problem should be stated more clearly to readers. The literature review is insufficient. Some recently published papers should be included in the references list.
2.Based on the proposed idea and obtained results in this paper, the authors should be able to present some more descriptions in conclusion part, for example, further study direction.
3.In simulation section, more analysis and descriptions should be given to show the effectiveness of the developed method.
4.The robustness of the proposed control method should be described.

---

### Official Review · Reviewer_fmpt · 2024-08-23
**Neural networks adaptive time-varying formation control for quadrotor unmanned aerial vehicles with finite-time prescribed performance**

**Rating:** 7
**Confidence:** 2

**Review:**

This paper focused on the time-varying control of quadrotor unmanned aerial vehicles system with finite-time performance function. To address the unknown nonlinear disturbances, neural networks were introduced. In addition, a novel performance function was formulated specifically for finite-time control. There are some problems that should be replied. Comments for this submission are given as follows:
1.The practicability of this paper should be further clarified. Which practical systems can be described as the system (1)?
2.There are a large number of studies in the existing literature on both the prescribed performance control. Compared with existing literature, what progress has been made in PPC control in this article.
3.How to implement the designed scheme? An algorithm and a block diagram should be provided.
4.Section VI does not suggest effective open problems and future issues that could require further investigations.

---

### Official Review · Reviewer_j87P · 2024-08-23
**accept**

**Rating:** 7
**Confidence:** 3

**Review:**

This paper studies the time-varying formation control problem of  aerial vehicles  with finite-time prescribed performance under nonlinear disturbances , which demonstrated excellent performance. The theory is correct and can be accepted after responding the following comments.
(1)There are many typos and grammar errors. The authors should have a native English speaker or software packages to perform the editing check.
(2)The conclusion of the article suggests using the present perfect tense for description.
(3)In the introduction, it is not enough to state the current work. It should be expended and reconstructed.

---

### Decision · Program_Chairs · 2024-09-06

Accept (Oral)